# Genomic screening of allelic and genotypic transmission ratio distortion in horse

**Nora Laseca**[1]*, **Ángela Cánovas**[2], **Mercedes Valera**[3], **Samir Id-Lahoucine**[4], **Davinia I. Perdomo-González**[3], **Pablo A. S. Fonseca**[5], **Sebastián Demyda-Peyrás**[1], **Antonio Molina**[1]

1 Department of Genetics, University of Cordoba, Córdoba, Spain, 2 Center of Genetic Improvement of Livestock, Department of Animal Biosciences, University of Guelph, Guelph, Ontario, Canada, 3 Department of Agronomy, School of Agronomy Engineering, University of Seville, Seville, Spain, 4 Department of Animal and Veterinary Science, Scotland's Rural College, Aberdeen, Scotland, United Kingdom, 5 Department of Animal Production, University of Leon, León, Spain

* ge2lagan@uco.es

**Data Availability Statement:** All relevant data are within the paper and its Supporting Information files. The data underlying the results presented in the study are available from https://data.mendeley.

## Abstract

The phenomenon in which the expected Mendelian inheritance is altered is known as transmission ratio distortion (TRD). The TRD analysis relies on the study of the transmission of one of the two alleles from a heterozygous parent to the offspring. These distortions are due to biological mechanisms affecting gametogenesis, embryo development and/or postnatal viability, among others. In this study, TRD phenomenon was characterized in horses using SNP-by-SNP model by TRDscan v.2.0 software. A total of 1,041 Pura Raza Español breed horses were genotyped with 554,634 SNPs. Among them, 277 horses genotyped in trios (stallion-mare-offspring) were used to perform the TRD analysis. Our results revealed 140 and 42 SNPs with allelic and genotypic patterns, respectively. Among them, 63 displayed stallion-TRD and 41 exhibited mare-TRD, while 36 SNPs showed overall TRD. In addition, 42 SNPs exhibited heterosis pattern. Functional analyses revealed that the annotated genes located within the TRD regions identified were associated with biological processes and molecular functions related to spermatogenesis, oocyte division, embryonic development, and hormonal activity. A total of 10 functional candidate genes related to fertility were found. To our knowledge, this is the most extensive study performed to evaluate the presence of alleles and functional candidate genes with transmission ratio distortion affecting reproductive performance in the domestic horse.

## Introduction

The deviation from the expected Mendelian inheritance of alleles from heterozygous parents to offspring is known as transmission ratio distortion (TRD) [1, 2]. This phenomenon reveals locus-specific selection acting between the heterozygous parents and the offspring genotype [3]. The TRD phenomenon has been observed in a large number of organisms, including plants [4], humans [5] and animals [2, 6–9]. This transmission deviation can be caused by a broad range of biological mechanisms affecting gametogenesis, fertilization or embryogenesis

com/datasets/vyby8h4x9x/2 Laseca Garcia, Nora (2023), "genotypes trios", Mendeley Data, V2, doi:10.17632/vyby8h4x9x.2.

**Funding:** This study was financed by the AGL-2017-84217-P Research project from the Ministry of Economy, Industry and Competitiveness of the Spanish Government. N. Laseca is funded by a FPI grant (PRE 2018-083492) from the Ministry of Science, Innovation, and Universities of the Spanish Government. The funders had no role in study design, data collection and analysis, decision to publish, or preparation of the manuscript.

**Competing interests:** The authors have declared that no competing interests exist.

[10], including germline selection [11], meiotic drive [12], gametic competition [13], embryo or fetal lethality [13, 14], imprinting resetting error [15] and differential postnatal viability [16]. According to the phase in the life cycle which occurs, it may affect both the distortion pattern and the associated phenotypic [3]. Currently, it is a challenge to determine the physiological stage at which TRD occurs and the associated biological cause in most species.

The development of new statistical methodologies using Bayesian approach allows to identify novel genomic regions with TRD [17, 18] elucidating more accurately the genetic mechanisms underlying the distortion on the Mendelian segregation. These new methodologies are based on tracing the allele inheritance from parents to offspring using genotypes of parent-offspring trios. Two parameterization models exist. The allelic parameterization allows to identify stallion- and mare-specific TRD, in addition to the overall TRD [6, 7]. Alternatively, genotypic parameterization models the interactions between alleles in the offspring genotypes, including the additive and dominance components of the TRD [2, 19]. These models allow comprehensive characterization of TRD across the genome by capturing all types of TRD and enables to anticipate the biological inheritance and the origin of TRD [20]. Even with these advances, knowledge about the possible causes and magnitude of TRD in livestock species is still largely unknown [18, 19, 21–23]. Especially in the case of the horse, the studies on the TRD phenomenon are still scarce [24] compared to other livestock species.

Nowadays, the increasing availability of high-throughput genotyping and sequencing technologies has provided a powerful source of genomic data and thus parents-offspring (stallion–mare–offspring) genotyped trios, allowing to study TRD signals in the genome. Using the TRD approach will allow us to identify genetic factors that regulate gamete, zygote, and embryo survival, and discover chromosomal regions with TRD that may contain genes with causal mutations affecting reproduction in horses. These findings could improve reproductive success in the equine sector. Notice that the reproductive efficiency of a livestock herd plays a crucial role in its profitability, and this is true for all livestock species, including horses. However, despite its significance, reproductive performance studies in equines are very scarce [25, 26] and lower reproductive performance, mainly due to its relative low fertility, has been reported [27]. In addition, although it is a trait of low heritability [28], it is still a selection target in most livestock species. Incorporating fertility traits into equine breeding programs remains a challenge, largely due to the difficulty of establishing reliable selection criteria. Furthermore, in many equine breeds, fertility traits are not given the same importance as other functional, behavioral, and morphological traits. In the case of the Pura Raza Español (PRE) breed horse, there is an interest in increasing the reproductive performance of its mares [28].

Therefore, the aim of this study was to investigate TRD in the horse genome to identify genomic regions showing altered Mendelian segregation deviations and to perform functional analysis of candidate genes located within the TRD regions to uncover affected biological processes and metabolic pathways associated with economically important traits such as reproduction in horses.

## Materials and methods

### Ethics statement

All experiments were performed in accordance with the guidelines in EU Directive 2012/63/EU. Blood and DNA samples were provided by the Asociación Nacional de Criadores de Caballos de Pura Raza Española (ANCCE). All owners gave their written consent to ANCCE for the use of their horse's DNA sample for scientific research. No animal experiments were carried out.

## Animal material, genotype data and imputed genotypes

A total of 1,041 Pura Raza Español breed horses were genotyped. Among them, 277 horses genotyped in trios (offspring-stallion-mare) were considered to perform the TRD analysis, among them, 154, 47 and 93 were offspring, stallions, and mares, respectively.

Genomic DNA was isolated from blood or hair samples using a DNeasy Blood & Tissue extraction kit (Qiagen). Horses were genotyped with high density Axiom™ Equine SNP Genotyping Array (Thermofisher), including 670,804 markers uniformly distributed across the entire genome [29]. The raw genotype data was analyzed following the "Best Genotyping Practices Workflow" with the Axiom Analysis Suite 5.0 software with the default parameters (DishQC $\geq$ 0.82 and call rate $\geq$ 0.97). Quality control analyses were performed using PLINK v1.9 software. Non-autosomal SNP markers and those with a call rate below 95% were discarded from further analysis. The final genomic data included 554,634 SNPs distributed across the 31 chromosomes.

To assess and reduce the possible effects of genotyping error on TRD estimation, the data were phased, and missing SNP were imputed using FImpute software [30].

## Analytical models of transmission ratio distortion

To determine SNPs subject to TRD across the whole horse genome a Bayesian approach to analyze TRD SNP-by-SNP was implemented [17]. Three models were considered to trace allele inheritance from parents to offspring in this population of horses. Two allelic models with parent-unspecific and -specific TRD effects [6] and a genotypic model with additive and dominance TRD effects were used [6].

**Allelic model.**   As described Casellas et al. [6, 7], the probability of allele transmission (P) from heterozygote parents (A/B) to offspring was parameterized including one overall TRD effect ($\alpha$) on a parent-unspecific model or differentiating between stallion- ($\alpha_s$) and mare-specific TRD effects ($\alpha_d$) on a parent-specific model:

$$P(A) = 1 - P(B) = 0.5 + \alpha \text{ and } P(B) = 1 - P(A) = 0.5 - \alpha,$$

$$P_i(A) = 1 - P_i(B) = 0.5 + \alpha_i \text{ and } P_i(B) = 1 - P_i(A) = 0.5 - \alpha_i \text{ with } i = [s \text{ U } d]$$

where; $\alpha$, $\alpha_s$ and $\alpha_d$ are TRD parameters which assumed flat priors within a parametric space ranging from -0.5 to 0.5.

**Genotypic model.**   As developed Casellas et al. [2], genotypic parameterization can be modeled by assuming additive ($\alpha_g$) and dominance ($\delta_g$; or over- / under-dominance) parameters, regardless of the origin of each allele. Following Casellas et al. [19], the probability of the offspring ($P_{off}$) from heterozygous-by-heterozygous mating are:

$$P_{off}(AA) = \frac{(1 + \alpha_g - \delta_g)}{4}, P_{off}(AB) = \frac{1 + \delta_g}{2} \text{ and } P_{off}(BB) = \frac{(1 - \alpha_g - \delta_g)}{4}$$

where; $\alpha_g$ and $\delta_g$ are additive- and dominance-TRD parameters, respectively.

For heterozygous-by-homozygous mating, correction for overall losses of individuals in terms of genotypic frequency are needed to guarantee $P_{off}(AA) + P_{off}(AB) + P_{off}(BB) = 1$. Thus, genotypic frequencies in offspring from AA × AB mating as example become:

$$P_{off}(AA) = \frac{(1 + \alpha_g - \delta_g)}{2x(1 + \alpha_g/2)}, P_{off}(AB) = \frac{(1 + \delta_g)}{2x(1 + \alpha_g/2)} \text{ and } P_{off}(BB) = 0$$

Flat priors were assumed for both $\alpha_g$ and $\delta_g$ within a deepened parametric space. The latter

range initially [−1, 1] for $\alpha_g$ with a $p(\alpha_g) = 1/2$ and becomes restricted to $[-1+ \delta_g, 1- \delta_g]$ with a $p(\alpha_g) = 2 / (2-2 \times \delta_g)$ when $\delta_g > 0$. For the $\delta_g$ component, the parametric space ranges $[-1, |\alpha_g|]$ with a $p(\delta_g) = 1/ (1 + \alpha_g)$.

**Statistical analyses.** The TRDscan v.2.0 software [17] was used to analyze TRD SNP-by-SNP across the whole horse genome. Both imputed and raw data were tested in order to assess and reduce the possible effects of genotyping error on TRD estimation. Each model was analyzed by running a Markov chain Monte Carlo with 110,000 iterations where the first 10,000 iterations were discarded as burn-in. To avoid false TRD, only SNPs with a minimal number of 10 informative offspring and two informative stallions and/or mares were analyzed as described by Id-Lahoucine et al. [17]. The statistical significance of the TRD was tested using a Bayes factor (BF) [31]. Both allelic and genotypic parameterizations were compared using the deviance information criterion (DIC, [32]) to determine the goodness-of-fit and the inheritance pattern of each SNP.

## Gene annotation and functional analysis

Genetic markers with significant TRD values were used to perform functional enrichment analysis. Genes were annotated within 250 kilobase pair (Kb) interval upstream and downstream from the SNP position using the Ensembl BioMart tool [33] with the *Equus caballus* reference genome (EquCab3.0. http://www.ensembl.org/Equus_caballus/Info/Index).

The genes annotated within the significant TRD regions were used to perform functional analysis including gene ontology (GO) analysis, metabolic pathway analysis and gene networks. The three GO categories (biological process (BP), molecular function (MF) and cellular component (CC)) were analyzed as described by Cánovas et al. [34]. The GO enrichment analysis and the metabolic pathways analysis were performed using AmiGO 2 [35], PANTHER GO-slim [36] and the Database for Annotation, Visualization and Integrated Discovery (DAVID) [37]. Significance levels were computed following a modification of Fisher's exact test. Multiple testing-corrected P-values were also obtained using the Benjamini and Hochberg algorithm, and only GO terms with Benjamini-corrected P-values 0.05 were considered.

## Results

### TRD signals in the horse genome

The presence of TRD was evaluated in the horse genome using two models (allelic and genotypic parametrization). To minimize genotyping errors that may generate false-positive TRDs, we compared all TRD regions using two datasets, the raw genotypes and the imputed genotypes. Most of them exhibited similar patterns, being 98.52% of TRD signals identical. A 3.73% of regions were significant when raw data were analyzed and showed null TRDs when data were imputed, suggesting that possible genotypic errors were corrected. Therefore, the imputed data were kept for further analysis following Id-Lahoucine et al. [17].

Regions with TRD signals were observed within the horse genome. With a threshold > 100 for BF according to Jeffreys' scale [38], a large number of SNPs (473) with TRD were found to exhibit decisive evidence of distortion. To target the most relevant TRD regions, a more restricted criteria were applied following Id-Lahoucine et al. [17]. For TRD regions, a minimal informative offspring (> 20) was considered to minimize possible false TRD. The approximate empirical null distribution of TRD at < 0.1% margin error was used in order to eliminate TRD generated by chance [20]. Finally, only SNP with moderate to strong TRD magnitude (i.e., |TRD|> 0.2) were considered. The prevalence of TRD was widely distributed across the horse genome (Fig 1), being the chromosomes ECA1, ECA2, ECA3, ECA4, ECA6, ECA12, ECA15, ECA18, ECA20, ECA22, ECA28 that higher presence of TRD.

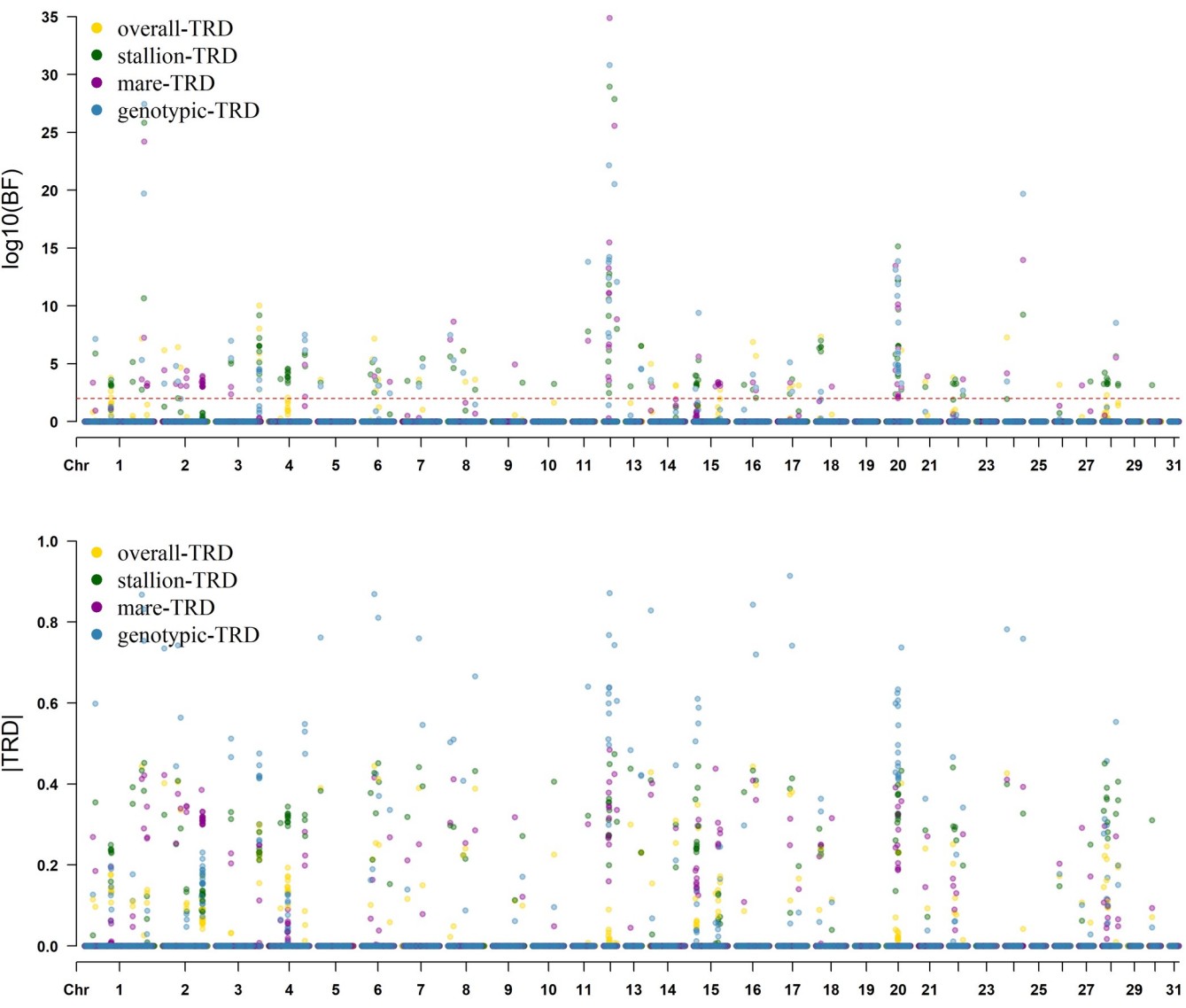

**Fig 1. Manhattan plot of SNPs with allelic and genotypic transmission ratio distortion across the horse genome.** A) Bayes Factor of SNPs with TRD. B) Magnitude of TRD. Overall-TRD (Yellow); stallion-TRD (Green); mare-TRD (Purple); genotypic-TRD (Blue). Suggestive line (red) in log10(BF) = 2 (decisive evidence).

**Allelic TRD patterns.** Most of the regions (77%) fitted better with the allelic model when comparing different TRD models based on DIC units. Thus, 140 SNPs were detected with the allelic TRD pattern showing decisive evidence (BF > 100) (S1 Table). Among them, 31 SNPs with TRD were supported by large evidence according to BF ($\geq 10^5$). The SNP (3: 117,731,559) with the largest evidence according to BF ($\geq 10^{10}$) exhibited a |TRD magnitude| of 0.3 and 121 informative offspring. The top (24) significant SNP markers according to BF are shown in S1 Table. In addition, SNPs with TRD were grouped according to the allele transmitted preferentially based on parental origin (stallion/mare). Allelic TRD from the stallion was detected in 63 SNPs and allelic TRD from the mare in 41 SNPs whereas 36 SNPs were with overall TRD. The observed overall TRD, stallion-TRD and mare-TRD had strong magnitudes (absolute values) of up to 0.44, 0.47 and 0.44, respectively.

On the other hand, SNPs physically linked were found showing the same TRD patterns, thus supporting the relevant of TRD region in the particular segment (S1 Table). These regions were located on different chromosomes, highlighting the regions on ECA1, ECA2, ECA3, ECA4, ECA15, ECA18, ECA20 and ECA28. Regions with mare-TRD effect covering several physical linked SNPs were found on chromosomes 1 (from 168,787,353 bp to 168,881,002), 2 (from 64,444,863 bp to 64,728,073 bp and from 107,779,387 bp to 108,134,857 bp), and 15 (from 65,988,635 bp to 66,617,913 bp) with |mare-TRD| ranging from 0.245 to 0.345, while regions with a stallion-TRD effect were located on ECA1 (from 71,520,651 bp to 71,678,688 bp), ECA4 (from 50,222,982 bp to 50,726,125 bp), and ECA15 (from 7,428,750 bp to 8,065,247 bp) with |stallion-TRD| ranging from 0.231 to 0.39. Finally, a parent unspecific TRD was noticeable on chromosomes 3 (from 117,722,958 bp to 117,793,513 bp), 18 (from 13,762,225 bp to 13,831,055 bp) and 20 (from 32,414,941 bp to 32,416,535 bp) with |overall-TRD| ranging between 0.212 and 0.3.

**Genotypic TRD pattern.** Based on the genotypic parametrization model and according to DIC value, 42 SNPs were detected showing a genotypic TRD pattern with additive- and/or dominance-TRD effects (BF > 100) (S1 Table). According to the statistical significance, 34 SNPs with TRD exceeding a $BF \geq 10^5$ were detected and 18 of them with $BF \geq 10^{10}$. The top significant SNPs according to BF are shown in S1 Table. The genotypic-TRD of the significant SNPs had strong magnitudes (absolute values) of up to 0.915. In addition, physically linked SNPs showing the same TRD patterns were found. These SNPs were found on ECA4 (from 97,130,353 bp to 97,219,055 bp), ECA12 (from 13,735,842 bp to 14,260,304 bp) and ECA20 (from 31,427,405 bp to 31,601,086 bp). The |genotypic-TRD| of these regions was 0.299 between and 0.768.

## Functional enrichment analysis

Functional annotation of positional candidate genes was performed for the 182 SNPs detected displaying transmission ratio deviations: 140 SNPs with allelic TRD (overall TRD, stallion- and mare-TRD) and 42 SNPs with genotypic TRD (additive- and dominance-TRD). A total of 296 genes were annotated in an interval of 500Kb (250 Kb up- and downstream) for the individual SNPs with allelic TRD effect (S2 Table). Among them, 30 genes were located overlapping the position of candidate SNP. In addition, 138 genes were annotated in an interval ± 250Kb for the SNPs with genotypic TRD effect finding 5 genes in the position of SNP (S3 Table).

**Gene ontology terms and metabolic pathways enriched in the list of candidate genes with allelic and genotypic TR.** Candidate genes annotated (296) in the interval of 500Kb for the individual SNPs showing allelic TRD were used to perform GO analysis including the three main GO categories of biological process, cellular component, and molecular function and into the respective metabolic pathways using different software. As results, 15 BP, 10 MF and 6 CC were significantly identified (p-value < 0.05; S4 Table). In turn, GO terms were clustered using PANTHER GO Slim as shown in Fig 2A and S1 Fig.

The top significant biological processes included magnesium ion homeostasis (GO:0010960) collagen catabolic process (GO:0030574) and potassium ion transmembrane transport (GO:0071805). The most significant GO terms for MF were NADP binding (GO:0050661), N,N-dimethylaniline monooxygenase activity (GO:0004499), and ATPase activity (GO:0016887). Finally, intracellular membrane-bounded organelle (GO:0043231), membrane raft (GO:0045121) and extracellular region (GO:0005576) were identified as top significant CC GO terms. On the other hand, 9 metabolic pathways were identified significantly (p-value <0.05, S4 Table). The top significant metabolic pathways were mTOR signaling pathway, taurine and hypotaurine metabolism and cytokine-cytokine receptor interaction.

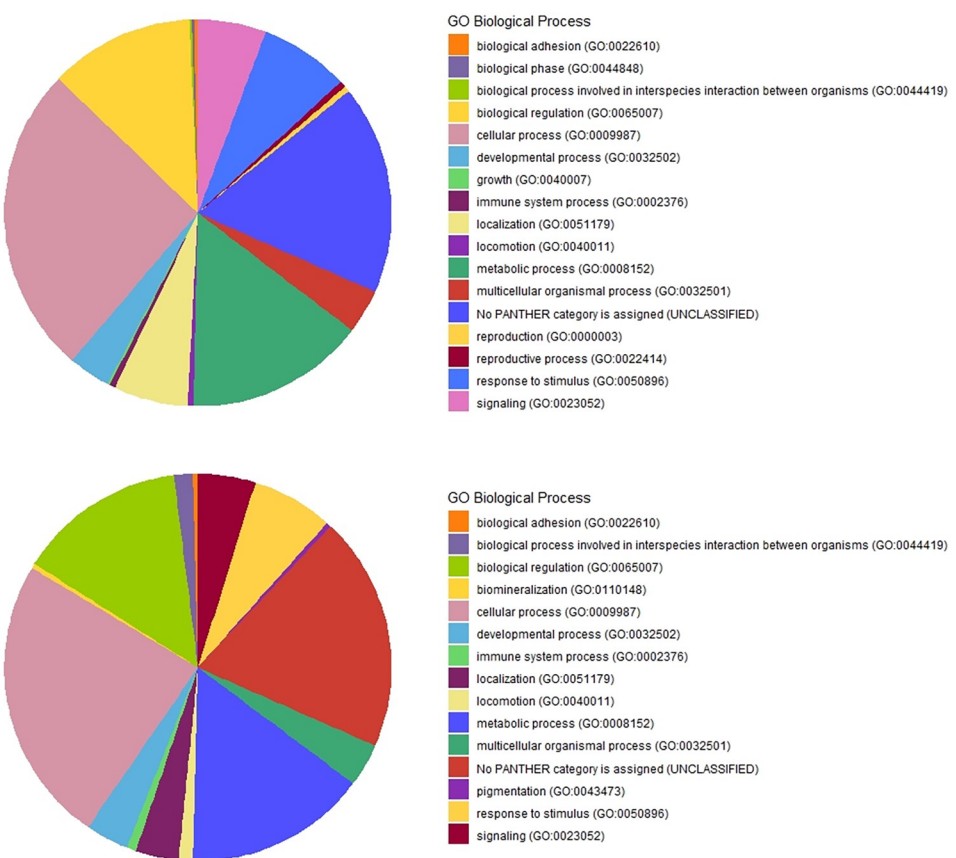

**Fig 2. Gene Ontology (GO) analysis using the list of genes with TRD effect in horse using AmiGO and PANTHER GO-Slim.** A) Biological Process sector diagram of the list of genes with allelic TRD. B) Biological Process sector diagram of the list of genes with genotypic TRD.

Regarding SNPs with genotypic TRD effect, Gene Ontology analysis was performed using the list of 138 genes annotated in the interval of 500Kb. The ontological analysis identified 22 BP, 16 MF and 2 CC significant enriched with p-value < 0.05 (S5 Table). Moreover, these GO terms were clustered using PANTHER GO Slim as shown in Fig 2B and S2 Fig.

The top significant biological processes included histone citrullination (GO:0036414), sensory perception of smell (GO:0007608) and negative regulation of actin filament polymerization (GO:0030837). Likewise, protein-arginine deiminase activity (GO:0004668), protein tyrosine/serine/threonine phosphatase activity (GO:0008138), and protein heterodimerization activity (GO:0046982) were identified as top significant molecular function GO terms. Similarly, the most significant GO terms of cellular component were nucleosome (GO:0000786) and nucleus (GO:0005634). Lastly,9 significant metabolic pathways were found (p-value < 0.05, S5 Table). The most significant metabolic pathway included neutrophil extracellular trap formation.

**Functional candidate genes.** A total of 11 functional candidate genes related to fertility and reproduction process were found (Table 1). Among them, 7 were associated with allelic TRD (*SPATA5*, *SPIRE1*, *PDPK1*, *BAG6*, *HSPA1L*, *EHMT2*, and *MSH5*) and 4 with genotypic TRD (*FGF8*, *NOS2*, *RELA*, and *PDGFB*). These genes are related to spermatogenesis, oocyte division, embryonic development, and hormonal activity, such as gonadotropin-releasing hormone, and testosterone biosynthesis.

Table 1. Functional candidate genes identified in the regions of SNPs with TRD effect related to fertility and reproductive processes.

| SNP | ECA | Gene | Start gene (bp) | End gene (bp) | Related to | Effect TRD |
|---|---|---|---|---|---|---|
| Affx-102676354 | 1 | FGF8 | 28,438,020 | 28,444,877 | regulation the ontogenesis of gonadotropin-releasing hormone neurons | genotypic TRD |
| Affx-102029138 | 2 | SPATA5 | 105,297,868 | 105,607,119 | spermatogenesis, sperm maturation and fertilization | allelic TRD |
| Affx-101211136 | 8 | SPIRE1 | 40,539,360 | 40,796,466 | oocyte division | allelic TRD |
| Affx-101258074 | 11 | NOS2 | 42,217,308 | 42,253,897 | regulation of female reproduction | genotypic TRD |
| Affx-102417813 | 12 | RELA | 29,516,708 | 29,556,026 | endometrosis | genotypic TRD |
| Affx-101805908; Affx-101457011 | 13 | PDPK1 | 41,420,452 | 41,494,281 | Survival of primordial follicles and activation of growing follicles. | allelic TRD |
| Affx-102696704; Affx-102783805; Affx-101513893; Affx-101938833; Affx-101714029 | 20 | BAG6 | 32,273,066 | 32,283,378 | spermatogenesis, maintaining testicular cell survival and testosterone biosynthesis | allelicTRD |
| | 20 | HSPA1L | 32,415,042 | 32,416,967 | sperm and block fertilization | allelic TRD |
| | 20 | EHMT2 | 32,472,911 | 32,485,948 | spermatogenesis and embryogenesis | allelic TRD |
| | 20 | MSH5 | 32,366,937 | 32,382,268 | primary ovarian insufficiency and meiotic arrest at stage IV of the spermatogenic cycle | allelic TRD |
| Affx-102804338 | 28 | PDGFB | 37,203,935 | 37,221,923 | modulation the primordial to primary follicle transition | genotypic TRD |

SNPs = single-nucleotide polymorphisms; ECA = equine chromosome; bp = base pairs; TRD = transmission ratio distortion

## Discussion

### Deviation from the expected mendelian inheritance on horse genome

This study focused on discovering genomic regions which are not transmitted according to the rules of Mendelian heritage but deviate from inheritance expectations showing signals of TRD in the horse genome. The analysis of the TRD phenomenon through genotyped trios allows us to accurately characterize the incidence, the biological nature, and the magnitude of TRD across the genome. As well as discovering functional genes related to reproductive mechanisms and fertility in the regions affected by TRD. For this purpose, of the 1,041 genotyped horses belonging to the Pura Raza Español horse breed, 277 PRE horses forming stallion-mare-offspring trios were used. The TRD SNP-by-SNP approach was performed on two datasets, the raw and imputed genotypes, to minimize genotyping errors that could potentially produce significant artifacts in the TRD analyses [20]. Some studies have pointed out that genotyping errors are a major problem in TRD analysis [5, 39]. Recently, Id-Lahoucine et al. [17] demonstrated that providing more informative and accurate data by imputation avoids potential TRD false positives generated by genotyping error. Based on our results, we demonstrated that using the imputed genotypes minimized the occurrence of false positive TRDs.

To date, the presence of TRD in the horse genome has been slightly demonstrated [24]. Our results demonstrated the existence of allelic and genotypic distortion patterns in the Pura Raza Español horse genome. Interestingly, SNPs were found close to each other with the same TRD patterns, evidencing the presence of important candidate mutations with different physically linked SNPs. Our results showed regions where multiple nearby SNPs exhibited the same distortion, providing support for the TRD effect. It is crucial to note that the magnitude of TRD for SNPs within these regions determines the intensity with which an allele is either over- or under-transmitted to subsequent generations [17]. Different magnitudes of TRD indicate varying levels of penetrance and distinct degrees of linkage disequilibrium between the SNPs.

As a result, regions displaying strong TRD were found to be associated with SNPs that exhibited high linkage disequilibrium with the underlying causal mutation. However, characterizing genomic regions with TRD in horses is indeed a challenging task, particularly when compared to species such as humans or cows. This difficulty arises due to the incomplete annotation of the equine genome. Unlike cattle, there are fewer studies available that report on alleles, haplotypes, mutations, and lethal genes [20, 40–43]. Consequently, the search for alleles and lethal mutations within our regions, where SNPs exhibit the same TRD effect, has proven to be complex thus far.

To ensure that our observed results are statistically significant, a Bayer Factor analysis was performed [31]. In TRD analysis, this analysis measures the change in the odds in favor of the model including TRD relative to the model with null TRD [6]. It is considered a measure of the significance of the detected TRD, given its ability to simultaneously combine the magnitude of the TRD and the sample size of informative offspring [17]. Our results showed statistical evidence in favor of the model including TRD indicating decisive evidence (BF > 100) according to [38] scale. In the allelic model, we observed that SNPs exhibiting both TRD (parent-specific and -unspecific TRD) showed higher BF values for overall TRD than for stallion-and/or dam-TRD. This might be because the detection of TRD is affected by the proportion of informative parents, this is heterozygous stallions and mares [20]. On the other hand, 7 of the 36 SNPs with overall TRD exhibited mare-TRD effect and 8 displayed stallion TRD effect. This demonstrates that overall TRD can also be captured as parent-specific TRD when sufficient informative data are available for the specific parent. Remark that the model with parent-unspecific TRD combines both sources of information (from stallions and mares) to estimate an overall TRD. In addition, this may confirm the parental origin of TRD even though it was identified as overall TRD. Previously, some studies have already reported the importance of sex dependence in TRD on other species [7, 21, 44], pointing out that some biological mechanisms causing TRD may be limited to one of the parental genders. Therefore, stallion-TRD could provide evidence of TRD linked to factors influencing sperm or, alternatively, phenomena of paternal impression. Likewise, mare-TRD could be linked to factors that affect ova fertility or maternal imprint [18, 19].

It is important to mention that implementing and comparing different parameterizations (genotypic and allelic) to capture all types of TRD [19], allows differentiating between the diverse potential causes of TRD, from gamete formation in the parental generation to the viability of the offspring. These parameterizations provide a tool to infer the biological source of TRD, linking it to haploid (allelic) or diploid (genotypic) reproductive stages. Based on goodness-of-fit in terms of DIC, we compared the different TRD models. Models with lower DIC values indicated a better fit and were considered statistically significant when the differences between models were greater than 3 DIC units. Our results showed that a greater number of regions fitted better to the allelic pattern than to the genotypic pattern. On the other hand, allelic patterns could also be considered as an additive effect in the genotype of the offspring, where the presence of the allele may cause a dosage effect and reduce the viability of the carrier offspring [20]. The presence or absence of a specific allele independently of the homologue may be sufficient to induce lethality and, as a result, generate TRD. We observed that certain regions with allelic patterns showed an almost complete absence for homozygous individuals, but also an underrepresentation of heterozygous offspring. Khatib et al. [45] and Id-Lahoucine et al. [20] already observed these distortion patterns in their studies on reproduction in candidate regions and whole-genome cattle, respectively. As for the genotypic pattern, no regions with classical recessive patterns were observed. Only heterosis (under- or over- representation of heterozygous offspring) patterns were observed.

## Functional candidate genes identified in the regions with deviations from mendelian expectations in horse

Functional annotation of candidate genes was performed for the 182 SNPs detected displaying deviations from Mendelian inheritance on the autosomal horse chromosomes. A total of 296 genes were annotated in regions with allelic TRD and 138 genes in regions with genotypic TRD. These positional genes located in TRD regions were examined to in deep investigate possible functional and molecular causes for the observed TRD. Functional analysis in both allelic- and genotypic-TRD regions revealed significant functional categories and biological processes related to histones, nucleosome and chromatin ("histone citrullination"; "heterochromatin assembly"; "nucleosome assembly"; "nucleosomal DNA binding") in which several genes were involved. The role of histones in spermiogenesis is well known, a defect in histone substitution or a modification could cause male infertility with azoospermia, oligospermia or teratozoospermia [46]. In turn, histone post-translational changes may have a major impact on chromatin structure and gene expression in the developing embryo [47]. Another functional category that plays a fundamental role in the development, behavior and reproduction of animals is that related to odor perception ("sensory perception of smell"; "odorant binding"; "olfactory receptor activity") [48] [49, 50] where several genes are involved. Interestingly, significant functional categories related to the immune system also appeared ("positive regulation of interleukin-6 production"; "innate immune response"). The immune system plays a critical role in fertilization, in the process of implantation of the embryo in the uterus, in the establishment and maintenance of pregnancy as well as in the testicular mechanism [51, 52]. Finally, another significant biological process related to reproduction was lipid metabolism ("lipid transporter activity"; "fatty acid omega-oxidation"). Different studies have suggested that an abnormal high-density lipoproteins metabolism hinders female fertility [53, 54] and that cholesterol and lipid homeostasis is important for male fecundity [55]. The genes associated with all these processes are mapped in regions where different TRD patterns were identified.

A total of 11 functional candidate genes related to fertility were found. Among them, 7 were associated with allelic TRD and 4 with genotypic TRD. The genes associated with female reproduction in regions with allelic-TRD were: *SPIRE1* and *PDPK1*.

The *SPIRE1* gene was related to the oocyte division [56]. Remarkably, this gene was identified in a genome-wide association study focused on mare fertility [57]. Another gene also found in this TRD study and in the association study on mare [57] was *PDPK1* gene. The phosphoinositide dependent protein kinase 1 gene has been associated with premature ovarian failure due to a massive primordial follicle activation in the knockout mouse [58].

Regarding the genes related to male fertility (*SPATA5*, *BAG6*, *HSPA1L*, *EHMT2*). The *SPATA5* is a member of the ATPase associated with diverse activities protein superfamily, which participates in mitochondrial morphogenesis during early spermatogenesis. This gene was first identified in mouse testis as a spermatogenesis-associated factor, which might participate in regulating mitochondrial structural integrity during spermatogenesis. It has been suggested that suppression of *SPATA5* expression leads to reduced expression of the vitellogenin gene, resulting in decreased fertility in males [59]. In addition, several studies have suggested that increased methylation of *SPATA5* may suppress its expression, which ultimately compromises sperm production and thus male fertility [60, 61]. Remarkably, *BAG6*, *HSPA1L* and *EHMT2* genes are associated with the same region with TRD, and all are related to male fertility. Regarding, BCL2-associated athanogene 6 (*BAG6*) plays a critical role in spermatogenesis by maintaining testicular cell survival [62] and in testosterone biosynthesis [63]. Another gene, heat shock protein family A member 1 like (*HSPA1L*) is testes-specific heat shock protein and is expressed strongly in the sperm. Some studies have shown that this specific protein can

result in antibody accumulation from previous infections and can effectively block fertilization [64]. However, recently, Wang et al. [65] revealed that *HSPA1L* is not essential for spermatogenesis, nor is it involved in heat-induced stress responses. Lastly, the *EHMT2* gene (also known as G9a) is a mammalian H3K9 methyltransferase, has an important role in mouse spermatogenesis and is essential for embryogenesis by transcriptional silencing [66, 67].

Finally, the gene *MSH5* related to male and female fertility was found. Mutations in *MSH5* gene have been previously described in the context of female infertility due to primary ovarian insufficiency [68]. But it was also described as a new gene in the context of male infertility [69], since previous studies showed that mutations in gene involved in recombination such as *MSH5* gene cause meiotic arrest at stage IV of the spermatogenic cycle.

On the other hand, genes found in regions with genotypic TRD were: *FGF8*, *NOS2*, *RELA* and *PDGFB* (female fertility). Fibroblast growth factor 8 (*FGF8*) regulates the ontogenesis of gonadotropin-releasing hormone neurons, which control the hypothalamus-pituitary-gonadal axis, and therefore reproductive success [70]. Many studies have shown that nitric oxide (NO) plays important role in female reproduction. Indeed, a study provided evidence that thyroid hormones dysregulation alters NOSs profiles, which suggested that NOSs/NO is possibly involved in the regulation of female reproduction [71]. Recent studies have reported the relation of nuclear factor kappaB (*NF-κB*) with endometrosis, a serious problem that mainly affects the fertility of older mares [72–74]. And one of the genes associated with a region with TRD is the *RELA* gene that belongs to the subunits *NF-κB*. Regarding, *PDGFB* (platelet-derived growth factor) gene has reported that modulates the primordial to primary follicle transition, which is essential for female fertility [75, 76].

## Conclusions

To our knowledge, this is the first extensive study to evaluate the presence of alleles with transmission ratio distortion in the domestic horse. Our results revealed deviations from Mendelian inheritance in 182 SNPs across the horse genome. Allelic (stallion- mare- parent-unspecific-TRD) patterns predominated over genotypic patterns. And functional analyses showed that the allelic and genotypic TRD regions identified in this study were associated with biological processes and molecular functions related to spermatogenesis, oocyte division, embryonic development, and hormonal activity. These findings contribute to a greater knowledge of TRD in the equine species and a better understanding of allele transmission distortion and could potentially be included in breeding programs with a major impact on the horse world. However, further studies with larger numbers of informative horse trios are needed to confirm these TRD patterns and validate the candidate genes in other equine breeds.

## Supporting information

**S1 Table. Significant SNPs identified with allelic transmission ratio distortion patterns by the allelic and genotypic model in horse genome.**
(XLSX)

**S2 Table. List of candidate genes annotated in an interval of 500Kb (250 Kb up- and downstream) for the individual SNPs with allelic TRD effect.**
(XLSX)

**S3 Table. List of candidate genes annotated in an interval of 500Kb (250 Kb up- and downstream) for the individual SNPs with genotypic TRD effect.**
(XLSX)

**S4 Table. Gene Ontology (GO) analysis (including the three main GO categories (biological process, molecular functions, and cellular components)) and metabolic pathways with DAVID database using the list of genes with allelic TRD in horse.**
(XLSX)

**S5 Table. Gene Ontology (GO) analysis (including the three main GO categories (biological process, molecular functions, and cellular components)) and metabolic pathways with DAVID database using the list of genes with genotypic TRD in horse.**
(XLSX)

**S1 Fig. Gene Ontology (GO) analysis using the list of genes with allelic TRD in horse using AmiGO and PANTHER GO-Slim.** Including A) molecular functions, and B) cellular components.
(ZIP)

**S2 Fig. Gene Ontology (GO) analysis using the list of genes with genotypic TRD in horse using AmiGO and PANTHER GO-Slim.** Including A) molecular functions, and B) cellular components.
(ZIP)

## Acknowledgments

The authors would like to thank the Asociación Nacional de Criadores de Caballos de Pura Raza Español (ANCCE) for their collaboration in this study and for providing the biological samples used in this study.

## Author Contributions

**Conceptualization:** Nora Laseca, Ángela Cánovas, Samir Id-Lahoucine, Antonio Molina.

**Data curation:** Nora Laseca, Mercedes Valera, Samir Id-Lahoucine, Antonio Molina.

**Formal analysis:** Nora Laseca, Ángela Cánovas, Samir Id-Lahoucine, Pablo A. S. Fonseca.

**Methodology:** Ángela Cánovas, Samir Id-Lahoucine.

**Resources:** Mercedes Valera, Davinia I. Perdomo-González, Sebastián Demyda-Peyrás.

**Supervision:** Ángela Cánovas, Samir Id-Lahoucine, Antonio Molina.

**Writing – original draft:** Nora Laseca, Ángela Cánovas, Antonio Molina.

**Writing – review & editing:** Nora Laseca, Ángela Cánovas, Mercedes Valera, Samir Id-Lahoucine, Davinia I. Perdomo-González, Pablo A. S. Fonseca, Sebastián Demyda-Peyrás, Antonio Molina.

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
