## [Decision Letter · Decision Letter 0]

31 May 2023

PONE-D-23-10812Genomic screening of allelic and genotypic transmission ratio distortion in horsePLOS ONE

Dear Dr. Laseca,

Thank you for submitting your manuscript to PLOS ONE. After careful consideration, we feel that it has merit but does not fully meet PLOS ONE’s publication criteria as it currently stands. Therefore, we invite you to submit a revised version of the manuscript that addresses the points raised during the review process.

We look forward to receiving your revised manuscript.

Kind regards,

Qinghua Shi

Academic Editor

PLOS ONE

Reviewers' comments:

Reviewer's Responses to Questions

**Comments to the Author**

1. Is the manuscript technically sound, and do the data support the conclusions?

Reviewer #1: Yes

2. Has the statistical analysis been performed appropriately and rigorously? 

Reviewer #1: Yes

3. Have the authors made all data underlying the findings in their manuscript fully available?

Reviewer #1: No

4. Is the manuscript presented in an intelligible fashion and written in standard English?

Reviewer #1: Yes

5. Review Comments to the Author

Reviewer #1: This study aimed to find regions with transmission ratio distortion (TRD) in the horse genome. The authors used data from 277 horses genotyped in offspring-stallion-mare trios at 670,804 markers distributed uniformly across the genome. By implementing allelic and genotypic TRD models of the software TRDscan, and filtering for stringent statistical criteria, the authors found 140 SNPs with allelic TRD and 42 SNPs with a genotypic TRD pattern. Functional enrichment analysis was performed for genes located in a 1Mb window around the identified TRD SNPs, and identified a significant enrichment of GO terms, such as spermatogenesis, gametogenesis, oocyte division, embryonic development and hormonal activity.

The research leading to this manuscript was performed properly and rigorously. The authors followed the protocol of the TRDscan software developers, and used stringent criteria for selecting candidate SNPs for TRD: SNPs with TRD pattern showing decisive evidence (BF>100), with <0.1% margin error, with at least 20 informative offspring and with |TRD| > 0.2. When conducting functional enrichment analysis, a Benjamini-corrected P-value cutoff of 0.05 was used.

Major revisions:

I would suggest that the authors put more emphasis on investigating the genomic characteristics of the regions with TRD, and test for patterns typically associated with TRD regions. Also, I suggest putting less emphasis on the functional enrichment analysis around the candidate SNPs.

In particular, the authors identified large TRD regions that contain multiple SNPs with similar TRD magnitudes. TRDs often involve multiple genes for their function, which they link together using inversions. It would be interesting to see if the identified regions contain inversions, have elevated polymorphism levels or lethal genes. One could obtain these parameters from the horse genome and its annotation. Also, is there a correlation of TRD strength and size of the region? A scatterplot would be informative of the relationship of these two, if there is correlation.

Figure 1 could include a panel b, which would be the same Manhattan plot as Figure 1, but with |TRD| on the y axis, like in Figure 4 of Id-Lahoucine et al. (2019).

The authors observed that SNPs with significant genotypic TRD show higher absolute TRD than those with allelic TRD. This comparison would be nice to see in a violinplot or a boxplot.

Table 1 and Table 2 list the top candidate SNPs with TRD, and they would be better suited for the Supplementary material, in my opinion.

The functional enrichment analysis is based on genes that are within 500 Kb of a TRD SNP. One concern is that 500 Kbs seem too inclusive, and one should maybe consider smaller windows. The other concern is that the authors end up with a total of 562 genes near SNPs with allelic TRD, of which one expects only a small proportion to be actually involved in TRD, the rest most likely just linked to the causal genes. Therefore, I think that the authors should less emphasis on the functional enrichment analysis and devote less space in the manuscript to reporting and discussing its results, as this part of the manuscript is too speculative. For example, the authors could include Table 3, which reports the candidate fertility and reproductive process genes, and choose one Venn diagram to show in the main text, and include the rest in the Supplementary material.

6. PLOS authors have the option to publish the peer review history of their article (what does this mean?). If published, this will include your full peer review and any attached files.

Reviewer #1: No

---

## [Author Response · Author response to Decision Letter 0]

19 Jun 2023

Dear editor,

we have followed your advice and comments: 

we have checked and corrected the style of PlosOne. We have added additional information in the Methods section: no animal experiments have been performed. Finally, we have added the dataset to a public repository and provided the URL code. In addition to the data and results already presented in the manuscript and in the additional information. 

Dear reviewer, 

we have followed your advice and comments to improve the manuscript.

Major revisions:

I would suggest that the authors put more emphasis on investigating the genomic characteristics of the regions with TRD, and test for patterns typically associated with TRD regions. Also, I suggest putting less emphasis on the functional enrichment analysis around the candidate SNPs.

Author Answer: The authors appreciate your opinion and considerations.

In particular, the authors identified large TRD regions that contain multiple SNPs with similar TRD magnitudes. TRDs often involve multiple genes for their function, which they link together using inversions. It would be interesting to see if the identified regions contain inversions, have elevated polymorphism levels or lethal genes. One could obtain these parameters from the horse genome and its annotation. Also, is there a correlation of TRD strength and size of the region? A scatterplot would be informative of the relationship of these two, if there is correlation.

Author Answer: Thank you for your comment. In order to identify inversions in TRD regions with SNPs with similar TRD magnitudes it would be necessary to sequence animals carrying these alleles in order to investigate the inversions. With array genotyping methodology it would only be possible to analyze copy number variation as we already did in another study [1].

On the other hand, regions annotated in the study with SNPs of similar TRD magnitude exhibit moderate levels of polymorphism (mean frequency is ~0.29). In particular these frequencies occur in SNPs with mare- or stallion- TRD where TRD occurs through only one single parent, while it is null through the other parent, resulting in maintenance of the level of polymorphism.

Regarding the search for lethal genes in these regions, we have not found any lethal mutation annotated in these regions, probably due to the insufficient annotation of the horse genome (so far) compared to the human or cow genome (Line 325-330).

Finally, it is important to note that we have not observed a generalized relationship between TRD strength and region size. Our findings indicate that there is no correlation in this regard. Instead, the magnitude of the TRD is influenced by the linkage disequilibrium of the SNPs present in the specific TRD region with the causal mutation. In other words, the strength of the TRD depends on how closely related the marker is to the causal mutation. When the marker is more closely related, it has the potential to capture a stronger TRD with a greater magnitude (Line 318-324).

Figure 1 could include a panel b, which would be the same Manhattan plot as Figure 1, but with |TRD| on the y axis, like in Figure 4 of Id-Lahoucine et al. (2019).

Author Answer: Thank you for your comment. The new Manhattan plot with |TRD| on the y axis have been included as panel b. (Line 158-159)

The authors observed that SNPs with significant genotypic TRD show higher absolute TRD than those with allelic TRD. This comparison would be nice to see in a violinplot or a boxplot.

Author Answer: Thank you for your comment. This is just a consequence of the implemented models which are based on different parameterizations. The parametric space of allelic parameterization ranges from -0.5 to 0.5, so the absolute maximum value is 0.5. In contract, the absolute maximum value for genotypic parameters is 1, as the parametric space of genotypic model is -1 to 1. This difference is because, form an allelic perspective, a parent could transmit one allele (e.g., A) with 0.5 (50%) and the opposite allele (e.g. B) with 0.5 (50%) under null TRD (TRD=0). If there is TRD, the probability of transmission of one particular allele could decrease up to 0 (0.5 - 0.5 = 0) or increase up to 1 (0.5 + 0.5 =1). Thus, the parameter space ranges from -0.5 to 0.5. In contrast, with the genotypic model, we do not focus on alleles but on the combination of alleles in the genotype and including additive and dominance TRD parameters, and the maximum probability of observing a particular genotype in the offspring generation is 1; thus, the parametric space ranges from -1 to 1 as described in Materials and Methods section. 

Table 1 and Table 2 list the top candidate SNPs with TRD, and they would be better suited for the Supplementary material, in my opinion.

Author Answer: Thank you for your comment. Following your suggestions, Table 1 and Table 2 have been suited for the Supplementary material. A new column has been added to S1 Table to mark the top candidate SNPs.

The functional enrichment analysis is based on genes that are within 500 Kb of a TRD SNP. One concern is that 500 Kbs seem too inclusive, and one should maybe consider smaller windows. The other concern is that the authors end up with a total of 562 genes near SNPs with allelic TRD, of which one expects only a small proportion to be actually involved in TRD, the rest most likely just linked to the causal genes. Therefore, I think that the authors should less emphasis on the functional enrichment analysis and devote less space in the manuscript to reporting and discussing its results, as this part of the manuscript is too speculative. For example, the authors could include Table 3, which reports the candidate fertility and reproductive process genes, and choose one Venn diagram to show in the main text, and include the rest in the Supplementary material.

Author Answer: Thank you for your comment. Following your suggestions, we have performed the functional enrichment analysis by reducing the search interval of genes around the SNPs showing TRD in ±250Kb upstream-downstream. In this way, thanks to their advice, we have focused on the genes with the strongest association to TRD, thus avoiding speculation. We have also reduced some of the information from the functional enrichment analysis by including the information in the Supplementary material.

1. Laseca N, Molina A, Valera M, Antonini A, Demyda-Peyrás S. Copy Number Variation (CNV): A New Genomic Insight in Horses. Animals. 2022;12(11). doi: 10.3390/ani12111435.

---

## [Decision Letter · Decision Letter 1]

11 Jul 2023

Genomic screening of allelic and genotypic transmission ratio distortion in horse

PONE-D-23-10812R1

Dear Dr. Nora Laseca,

We’re pleased to inform you that your manuscript has been judged scientifically suitable for publication and will be formally accepted for publication once it meets all outstanding technical requirements.

Kind regards,

Qinghua Shi

Academic Editor

PLOS ONE

Additional Editor Comments (optional):

Dear Dr. Nora Laseca,

Based on the suggestion of the reviewer and my thinking, I am pleased to tell you that your manuscript has been accepted for publish in this journal.

Reviewers' comments:

Reviewer's Responses to Questions

**Comments to the Author**

1. If the authors have adequately addressed your comments raised in a previous round of review and you feel that this manuscript is now acceptable for publication, you may indicate that here to bypass the “Comments to the Author” section, enter your conflict of interest statement in the “Confidential to Editor” section, and submit your "Accept" recommendation.

Reviewer #1: All comments have been addressed

2. Is the manuscript technically sound, and do the data support the conclusions?

Reviewer #1: Yes

3. Has the statistical analysis been performed appropriately and rigorously? 

Reviewer #1: Yes

4. Have the authors made all data underlying the findings in their manuscript fully available?

Reviewer #1: Yes

5. Is the manuscript presented in an intelligible fashion and written in standard English?

Reviewer #1: Yes

6. Review Comments to the Author

Reviewer #1: The authors addressed the issues and questions that I raised with regards to their manuscript. They have conducted the suggested analyses to further assess the genomic characteristics of the candidate regions causing TRD. They have also modified their figures, tables and the main text, as recommended. They have chosen more stringent criteria for their GO analysis, which makes their analyses and results more meaningful.

7. PLOS authors have the option to publish the peer review history of their article (what does this mean?). If published, this will include your full peer review and any attached files.

Reviewer #1: No

---

## [Editor Report · Acceptance letter]

1 Aug 2023

PONE-D-23-10812R1 

Genomic screening of allelic and genotypic transmission ratio distortion in horse 

Dear Dr. Laseca:

I'm pleased to inform you that your manuscript has been deemed suitable for publication in PLOS ONE. Congratulations! Your manuscript is now with our production department. 

Kind regards, 

on behalf of

Professor Qinghua Shi 

Academic Editor

PLOS ONE